# Virulence and Antimicrobial Resistance in Canine *Staphylococcus* spp. Isolates

**DOI:** 10.3390/microorganisms9030515

**Published:** 2021-03-02

**Authors:** Fabrizio Bertelloni, Giulia Cagnoli, Valentina Virginia Ebani

**Affiliations:** Department of Veterinary Sciences, University of Pisa, 56124 Pisa, Italy; g.cagnoli@studenti.unipi.it (G.C.); valentina.virginia.ebani@unipi.it (V.V.E.)

**Keywords:** *Staphylococcus*, dog, antimicrobial resistance, virulence, biofilm

## Abstract

Dogs are reservoirs of different *Staphylococcus* species, but at the same time, they could develop several clinical forms caused by these bacteria. The aim of the present investigation was to characterize 50 clinical *Staphylococcus* isolates cultured from sick dogs. Bacterial species determination, hemolysins, protease, lipase, gelatinase, slime, and biofilm production, presence of virulence genes (*lukS/F-PV*, *eta*, *etb*, *tsst*, *icaA*, and *icaD*), methicillin resistance, and antimicrobial resistance were investigated. Most isolates (52%) were *Staphylococcus pseudointermedius*, but 20% and 8% belonged to *Staphylococcus*
*xylosus* and *Staphylococcus chromogenes*, respectively. Gelatinase, biofilm, and slime production were very common characters among the investigated strains with 80%, 86%, and 76% positive isolates, respectively. Virulence genes were detected in a very small number of the tested strains. A percentage of 14% of isolates were *mecA*-positive and phenotypically-resistant to methicillin. Multi-drug resistance was detected in 76% of tested staphylococci; in particular, high levels of resistance were detected for ampicillin, amoxicillin, clindamycin, and erythromycin. In conclusion, although staphylococci are considered to be opportunistic bacteria, the obtained data showed that dogs may be infected by *Staphylococcus* strains with important virulence characteristics and a high antimicrobial resistance.

## 1. Introduction

The *Staphylococcus* genus includes a heterogeneous group of Gram-positive bacteria. The genus comprises 81 species and subspecies divided in the two groups, coagulase-positive (CoPS) and coagulase-negative (CoNS), based mainly on clinical and diagnostic aspects [1,2]. CoPS are well recognized as important human and animal pathogens, while the role of CoNS as primary pathogens or opportunistic bacteria is still under discussion [3,4]. Staphylococci are ubiquitous bacteria, and most of them are mammalian commensals that colonize niches such as skin, nares, and diverse mucosal membranes [1].

Animals are generally considered carriers of both CoPS and CoNS, mainly at skin level, and they are sometimes responsible to carry these bacteria, or their toxins, to humans via direct contact or foods [1,5,6]. However, staphylococci can also cause severe disease in livestock (mainly dairy animals) and pets [1,7].

Dogs play an important role in *Staphylococcus* epidemiology. Indeed, dogs are frequently carriers of CoPS and CoNS at the skin and mucous membrane levels. On the other hand, dogs can suffer by different clinical forms as a consequence of *Staphylococcus* opportunistic infections. Though skin and soft tissues are the most common sites of infection, any body system can be affected. Otitis and pyoderma are the main diseases associated with staphylococci, but wound or surgical site infections, urinary tract infections (UTIs), toxic shock syndrome, necrotizing fasciitis, arthritis and osteomyelitis, peritonitis, ocular infection, and septicemia can occur, too [8]. The coagulase-positive *Staphylococcus pseudointermedius* is the most detected species in both healthy and sick dogs, and it is assumed that it is host-adapted [9]. Though with a lower detection rate, other CoPS and CoNS can be isolated and cause diseases in dogs: *Staphylococcus aureus*, *Staphylococcus schleiferi*, *Staphylococcus epidermidis*, *Staphylococcus haemolyticus*, *Staphylococcus saprophyticus*, *Staphylococcus sciuri*, and *Staphylococcus warneri* [10,11,12]. The exact arrangement of different *Staphylococcus* species as saprophytic, commensal, opportunistic, or pathogens for dogs is still under evaluation. Many studies have been focused on the discovery of some bacterial or host factors that could be at the basis of disease development, even if a mix of these factors are probably involved [1,7,9,13]. Regarding the bacterial side, different virulence factors have been detected in strains isolated from dogs, in particular from skin lesions, including different types of exfoliative toxins, leucocidins, superantigens, enterotoxins, invasion enzymes, and biofilm producers [10,11,14,15,16].

Regardless of the ways of and reasons for infection development, a frequent issue is antibiotic treatment. Indeed, staphylococci are not outsiders to the antimicrobial resistance and multidrug resistance (MDR) problem [17]. Different authors have reported a decreased susceptibility to β-lactams, clindamycin, tetracyclines, and (less frequently) fluoroquinolones and trimethoprim-sulfamethoxazole [10,11,12,18,19,20]. Probably, the biggest alarm associated with *Staphylococcus* spp. in this context, is methicillin resistance. A few years after the introduction of this molecule for clinical treatment, methicillin-resistant *S. aureus* (MRSA) strains rose up and spread [21]. Though *S. aureus* strains are the principal bacteria involved in methicillin resistance, some other species, including *S. pseudointermedius*, can develop resistance to this antimicrobial [16,21,22].

Finally, the exact role of dogs as reservoirs and source of zoonotic staphylococci is under discussion [1,23]. However, in view of the One Health perspective, this aspect cannot be ignored because dogs are anthropized animals that often live in very close contact with their owners.

The aim of the present investigation was to characterize *Staphylococcus* strains isolated from sick dogs. Phenotypic and genotypic characters related to virulence properties, as well as antimicrobial, multidrug, and methicillin resistance, were evaluated.

## 2. Materials and Methods

### 2.1. Bacterial Strains, Isolation and Identification

All investigated bacterial isolates were obtained during routine diagnostic activity performed in the Laboratory of Infectious Disease of the Department of Veterinary Sciences, University of Pisa. No animals were specifically recruited for this study; all samples were collected by private veterinarians and sent to the laboratory in order to obtain a bacteriological diagnosis compatible to the observed clinical presentation. Samples included urine and swabs from the skin, ear, surgical site infections (SSIs), and nose. Swabs and urine pellets were initially streaked on Tryptose Blood Agar Base (Thermo Fisher Diagnostics S.p.A., Milan, Italy) supplemented with 5% sheep blood (blood agar) and incubated at 37 °C aerobically for 24 h. *Staphylococcus* spp. initial identification was carried out via colony morphology, Gram staining, catalase test and growth on mannitol salt agar (Thermo Fisher Diagnostics). All presumptive staphylococci were confirmed and identified at the species level with API STAPH^®^ (bioMérieux SA, Marcy l’Etoile France) following the manufacturers’ instructions. In order to discriminate between *S. aureus* and *S. pseudointermedius*, the protocol proposed by Sasaki et al. 2010 was employed [24]. All the strains were stored at −20 °C in brain heart infusion broth (BHI) (Thermo Fisher Diagnostics) supplemented with 25% glycerol fur further analyses. Staphylococcal isolates were cultured from January 2019 to February 2020; all strains included in the study were considered to be the main ones responsible for the clinical forms of the investigated dogs.

### 2.2. Phenotypic Characterization

All isolates were tested for hemolytic activity, proteases, lipases, and gelatinases production, as well as for slime and biofilm production. Hemolytic activity was evaluated by plating isolate pure cultures on blood agar incubated at 37 °C for 24 h; the presence of a clear zone around the colonies was interpreted as α-hemolysis, a wider zone of partial or incomplete clarification of the medium was interpreted β-hemolysis, and the absence of activity on red blood cells was recorded as γ-hemolysis [25]. For proteases production, isolates were cultured on casein agar (50 g of skim milk, 10 g of agar, and 1000 mL of distilled water), incubated at 37 °C for up to 14 days and checked daily for the presence of a clean zone (casein hydrolysis) around the colony. Gelatinase production was evaluated by the inoculation of isolates into tubes containing 4 mL of a gelatin medium (10 g of yeast extract, 15 g of triptone, 120 g of gelatine from bovine skin, and 1000 mL of distilled water); tubes were incubated initially at 30 °C for 7 days and successively, after 1 week, at 4 °C for at least 1 h. If bacteria produced gelatinase, the medium turned to liquid. Isolates were cultured in lipase test agar (10 g of tryptone, 10 mL of Tween 80, 0.111 g of CaCl_2_, 15 g of agar, and 1000 mL of distilled water) for 24/48 h at 37 °C in order to detect lipase activity; positive isolates produced a clear halo of salt precipitation around the colonies [25,26].

The Congo red agar (CRA) test was employed for the detection of slime production. Isolates were inoculated onto CRA plates (0.8 g of Congo red, saccharose of 36 g, 1000 mL of BHI, and 15 g of agar) and incubated for 24 h at 37 °C. Slime-producing strains formed black colonies, while non-producing ones produced red colonies; in particular, isolates were classified using a six-color scale ranging from very black (vb) to very red (vr) [27]. A crystal violet microtiter plate assay was employed for biofilm production following previously reported protocols, with slight modifications. Briefly, overnight bacteria cultures were diluted 1:200 in trypticase soy broth (TSB) (Thermo Fisher Diagnostics) with the addition of 1% glucose. The assay was performed in a flat-bottomed, tissue culture-treated polystyrene microtiter plates aerobically incubated at 37 °C for 24 h. After incubation, wells were rinsed and dried, and biofilms were stained with 0.1% crystal violet dye. Finally, after washing, the dye was resuspended with 30% glacial acetic acid, and the optical density at 490 nm (OD_490_) was measured. The biofilm assay was repeated in 3 independent experiments for each isolate. Biofilm production was evaluated with a 4-point scale: nonadherence (OD < ODcontrol), weak adherence (ODcontrol < OD < 2 ODcontrol), moderate adherence (2 ODcontrol < OD < 4 ODcontrol), and strong adherence (4 ODcontrol < OD) [15,26].

The disc diffusion method (EUCAST, The European Committee on Antimicrobial Susceptibility Testing, disk diffusion method for anti-microbial susceptibility testing version 6.0) was employed to determine the resistance to the following antimicrobials (Thermo Fisher Diagnostics): amikacin (AK; 30 µg), amoxycillin and clavulanic acid (AMC; 20–10 µg), amoxycillin (AML; 10 µg), ampicillin (AMP; 10 µg), cefalexin (CL; 30 µg), cephalothin (KF; 30 µg), cefotaxime (CTX; 30 µg), ceftazidime (CAZ; 30 µg), ciprofloxacin (CIP; 5 µg), clindamycin (DA; 2 µg), doxycycline (DO; 30 µg), enrofloxacin (ENR; 5 µg), erythromycin (E; 10 µg), gentamicin (CN; 10 µg), neomycin (N; 10 µg), rifampicin (RD; 30 µg), streptomycin (S; 10 µg), trimethoprim–sulfamethoxazole (SXT; 19:1, 25 µg), tetracycline (TE; 30 µg), and tobramycin (TOB; 10 µg). Antimicrobial resistance test was performed on Mueller–Hinton agar plates (Thermo Fisher Diagnostics) incubated at 35 °C for 16–20 h and interpreted according to breakpoints provided by EUCAST or CLSI (The Clinical and Laboratory Standards Institute) [28,29]. One isolate was considered multidrug-resistant (MDR) if it was resistant at least to one antibiotic in three or more different antimicrobial classes [30].

For methicillin resistance evaluation, oxacillin MIC was determined with the broth microdilution method [31]. *S. aureus* isolates were considered to be methicillin-resistant if an MIC value ≥ of 4 µg/mL was detected, while, for the other *Staphylococcus* species, the breakpoint was set at 0.5 µg/mL [32].

### 2.3. Genotypic Characterization

DNA was extracted from overnight cultures by a commercial kit, the Tissue Genomic DNA Extraction Kit (Fisher Molecular Biology, Trevose, PA, USA), following the manufacturer’s guidelines.

The presence of genes encoding for the following virulence factors was screened using primers and protocols previously described by other authors: Panton–Valentine leukocidine (PVL) (*lukS/F-PV*), exfoliative toxins a and b (*eta* and *etb*), toxic shock syndrome toxin (*tsst*), and intercellular adhesion (*icaA* and *icaD*) [33,34,35].

The *mecA* and *mecC* genes responsible for methicillin resistance were searched for using previously published primers and protocols [36,37].

## 3. Results

### 3.1. Bacterial Strains

Overall, 50 *Staphylococcus* spp. isolates were included in the study. Twenty isolates were cultured from skin swabs, 13 from ear swabs, eight from urines, seven from SSI swabs, and two from nose swabs.

The most detected *Staphylococcus* species was *S. pseudointermedius* (26 isolates), followed by *Staphylococcus xylosus* (10 isolates), *S. aureus* (six isolates), and *Staphylococcus chromogenes* (four isolates). One isolate was obtained for each of the following species: *Staphylococcus capitis*, *Staphylococcus simulans*, *S. haemolyticus,* and *Staphylococcus hyicus*. Table 1 shows the distribution of *Staphylococcus* species in relation to the site of infection.

### 3.2. Phenotypic Characterization

Most of isolates, 27/50 (54.0%), showed α/β hemolytic activity, while 13/50 (26.0%) and 10/50 (20.0%) showed β and α hemolytic activity, respectively. Regarding the other investigated enzymatic activities, 23/50 (46.0%) of the isolates produced proteases, 26/50 (52.0%) produced lipases, and 40/50 (80.0%) produced gelatinases.

Slime production was detected in 38/50 (76.0%) of isolates. In detail, 16/50 (32.0%) of the isolates were classified as “very black”, 13/50 (26.0%) as “black”, and 9/50 (18.0%) as “almost black”. Among negative isolates, 10/50 (20.0%) and 2/50 (4.0%) showed “red” and “very red” profiles, respectively. All but seven isolates were biofilm producers. Among them, 41/50 (82.0%) were weakly adherent and 2/50 (4.0%) moderately adherent.

Table 2 summarizes the results obtained by the phenotypic characterization.

Table 3 reports data regarding antimicrobial resistance. High levels of resistance were detected for amoxicillin (42/50), ampicillin (41/50), clindamycin (34/50), streptomycin (32/50), neomycin (31/50), erythromycin (31/50), and tetracycline (27/50). Most of the isolates were susceptible to amikacin (41/50), rifampicin (40/50), amoxycillin and clavulanic acid (30/50), and (in general) to cephems. A large number of isolates (38/50, 76.0%) were multidrug-resistant (Table 2). The remaining non-multidrug-resistant strains were two *S. pseudointermedius* strains isolated from the skin, two *S. pseudointermedius* strains isolated from the ear, one *S. pseudointermedius* strain isolated from an SSI, one *S. aureus* strain isolated from an SSI, one *S. chromogenes* strain isolated from the ear, and one *S. capitis* strain isolated from the ear.

Regarding methicillin resistance, 22/50 (44.0%) of isolates showed an oxacillin MIC over the breakpoint (Table 2).

### 3.3. Genotypic Characterization

None of the investigated isolates had the genes *lukS/F-PV*, *eta*, *etb*, and *mecC*.

Two out of fifty isolates (4.0%) were positive for *icaAD,* and 1/50 (2.0%) isolate was positive for *tsst* (Table 2); 12/50 (24.0%) of isolates had the *mecA* gene, and, among them, only seven *mecA*+ isolates showed a phenotypic resistance (four *S. pseudointermedius*, one *S. aureus*, one *S. xylosus,* and one *S. chromogenes* strains). On the other hand, 15 of the examined staphylococci that exhibited an oxacillin MIC greater than the breakpoint did not harbor *mec* genes (seven *S. pseudointermedius*, five *S. xylosus*, one *S. simulans*, one *S. haemolyticus,* and one *S. capitis* strain). Finally, five *S. pseudointermedius* isolates were *mecA*-positive but susceptible to methicillin.

## 4. Discussion

Staphylococci are important pathogens or opportunistic bacteria in dogs. In this study, a collection of 50 isolates from different sites of sick dogs were analyzed and characterized while taking different aspects in account.

First of all, each species was identified for each isolate. More than 50% of isolates were *S. pseudointermedius.* Indeed, this staphylococcal species is adapted to the Canidae family, the individuals of which usually serve as carriers [9]. *S. pseudointermedius* is an opportunistic pathogen, so it is very common that “pacific cohabitation” evolves in clinical infections [8]. However, other *Staphylococcus* species—both CoPS and CoNS—were isolated. *S. xylosus* and *S. chromogenes*—with 10 and 4 isolates detected, respectively—represented about 30% of total. These two coagulase-negative staphylococci are common bacteria of bovine and ovine milk flora, and they are rarely detected among dogs [35,38,39,40,41]. The human and animal pathogen *S. aureus* was detected in a few samples (12%); as confirmed by our data, dog is not considered a typical reservoir of this bacterial species [1,38]. The other detected staphylococcal species were poorly represented, confirming that they are rarely involved in canine infections and they probably can be better considered to be fortuitous rather than opportunistic pathogens.

Regarding the sites of infection, most of strains were cultured from the skin and ears. Staphylococci have a particular tropism for surface area of the body in mammals; they are part of the common flora of these anatomical districts, and this condition could often evolve in infections [42]. Many studies have reported *Staphylococcus* as a very frequent causative agent of canine dermatitis and otitis [20,38,40,43,44].

Percentages of 16% and 14% of isolates were found to come from UTIs and SSI, respectively. Staphylococci are considered to be frequently involved in canine UTIs, second only to Enterobacteriaceae [45,46,47]. In light of their opportunistic nature, it is not surprising that staphylococci could easily colonize surgical sites, as reported by other authors [48]. Our results confirmed and highlighted the importance of adequate antiseptic and prophylactic interventions to prevent or reduce SSIs.

The production of different enzymes or other molecules, defined as virulence effectors, represents some of the weapons that bacteria have in their arsenal for host invasion. In the present investigation, most of tested isolates were able to produce hemolysins, proteases, lipases, and gelatinases. *Staphylococcus* hemolysins are important virulence factors, with cytotoxic activity, that contribute to cell membrane damage and the lysis of keratinocytes. In particular, α hemolysin is a potent pore-forming cytolysin with a high-affinity for mammalian cells, while, β hemolysin is a sphingomyelinase that hydrolyzes plasma membrane lipids and has a lytic activity that is not as efficient as that of other hemolysins [49,50]. However, both hemolysins contribute to staphylococcal pathogenesis and they are typically associated with virulent *Staphylococcus* strains. All screened isolates were able to produce hemolysins; overall, 80% of isolates were α hemolytic and most of them (54% of total) were α/β hemolytic. Proteases, lipases, and gelatinases are extracellular enzymes that contribute to immune evasion and host tissue penetration, mainly facilitating movement through soft tissues like the epidermis and dermis [13,42]. Gelatinase production was a very common characteristic among tested bacteria, with 80% of positive isolates. Protease and lipase production was detected in about half of analyzed bacteria, suggesting these virulence factors were less spread among staphylococci involved in dog infections.

Slime and/or biofilm production offers a great advantage to bacteria. It promotes bacterial persistence in the environment, the colonization of biotic and abiotic surfaces, and protection from antimicrobial and disinfectants [51,52]. Moreover, previous studies have demonstrated that biofilm formation offers a great advantage to *Staphylococcus* during body colonization and infection, in particular of skin and wounds, and this characteristic is more often associated with pathogenic and invasive strains [15,53,54]. In accordance with these researches, most of tested isolates were found to be slime or biofilm producers in CRA or microplate assays; indeed, only two isolates scored negative on both tests (Appendix A). The microplate test was more suitable to detect this property; 10 isolates were positive in microplate assay and negative on CRA, whereas five isolates were positive on the CRA but did not produce biofilm on the microplate assay (Appendix A). Though 80% of isolates were positive in the microtiter assay, they did not show high levels of biofilm production; this result was in agreement with other studies carried out on staphylococci isolated from diseased dogs [26,53]. Generally, strains from foods, especially milk, are strong biofilm producers [55].

Few isolates were positive for the investigated virulence genes. In particular, the genes coding for PVL and exfoliative toxins, important factors in humans skin colonization and infection [33,34], were not detected. Though these virulence factors can be found in animal or environmental strains, they are generally associated with human clinical pathogens and rarely detected in staphylococci of veterinary origin [56,57,58]. For these reasons, it could be supposed that the investigated isolates were not of human origin. Only one *S. aureus* strain harbored the gene for the toxic shock syndrome toxin. This superantigen strongly favorites staphylococcal infection, but its detection in animal staphylococci is very variable [16,56,58,59]. In the present study, the investigated genes were selected on the basis of available information reported in recent publications, the site of infection, bacterial species, and zoonotic implications. However, in the future, it would be interesting to expand the set of virulence genes searched in order to better understand those involved in canine staphylococcal infections.

Considering the high number of isolates that produce biofilms detected in phenotypic assays, a high positivity to *icaAD* genes was expected. However, only two isolates were positive. The *ica* operon is considered to be the main operon responsible for biofilm formation in the *Staphylococcus* genus. However, some recent studies have highlighted that there is not an absolute correlation between the presence/absence of *ica* genes and biofilm production and the involvement of many other possible factors [26,41,51,53].

Methicillin resistance among *Staphylococcus* strains represents a serious problem of public health; animals could act as reservoirs and sources of methicillin-resistant staphylococci [21]. In this investigation, 24% of isolates harbored the *mecA* gene, while 44% were phenotypically resistant. Only seven isolates—four *S. pseudointermedius*, one *S. aureus*, one *S. xylosus,* and one *S. chromogenes* strains—showed a concordance between genotypic and phenotypic investigations. The detection of susceptibility to this antibiotic by five *mecA*-positive *S. pseudointermedius* strains could suggest the circulation of a defective antimicrobial resistance gene among staphylococci belonging to this species in the investigated canine population. This event was described for *mecA* gene, as well as for other genes [10,11,60]. On the other hand, 15 isolates (seven *S. pseudointermedius*, five *S. xylosus*, one *S. simulans*, one *S. haemolyticus*, and one *S. capitis*) were resistant despite both *mecA* and *mecC* not being detected. This evidence could have been due to the presence of other not investigated, less frequent methicillin-resistant genes or to a β-lactamase hyper-production, as also reported by other authors [10,21].

Finally, susceptibility to antimicrobial resistance was evaluated. Generally, a high level of antimicrobial resistance was detected, with a large number of MDR isolates (84%). This finding could have partially been due to the high number of molecules’ classes included in the study. Unfortunately, in the absence of international or national reference guides for antimicrobial surveillance, we included all the antimicrobial routinely tested for clinical purpose. On the other hand, pathogenic strains isolated form sick animals are often MDR, as also reported by other studies [10,11,18,19,53]. Resistance to antimicrobials could represent an advantage for bacteria in the evolution/progression of the disease, so it could be considered to be a virulence factor. In agreement with other studies, a low susceptibility was detected against penicillins, mainly ampicillin (82% of resistant isolates) and amoxicillin (84% of resistant isolates), molecules largely employed for many years [11,19,53]. A decreased susceptibility was found for clindamycin (68% of resistant isolates) and erythromycin (62% of resistant isolates), too. Recently, some studies, carried out on staphylococci isolated from dogs, reported a loss of the efficacy of these antimicrobials [10,19,53].

In our study, the most effective molecules were amikacin, rifampicin, and cephalosporins (Table 3). These results were in contrast with some investigations but in agreement with other studies. In particular, in *S. pseudointermedius* isolates collected from dogs with clinical pyoderma in Northern Italy, Meroni and colleagues detected a 100% susceptibility against amikacin, about 90% susceptibility against rifampicin, and a high level of resistance to β-lactams, which was in line with our results [53]. In a study on canine staphylococci from United Kingdom and Romania, Hiritcu et al. found variable levels of resistance to cefalexin, among isolates belonging to methicillin-susceptible *S. pseudointermedius*, methicillin-resistant *S. pseudointermedius*, and CoNS collected from the two countries, ranging from 0% to 57%; in the same way, tetracycline resistance ranged between 30% and 100%. Both results were similar to the ones obtained in our survey [10]. In a study conducted in Belgium by Jong et al., the detected susceptibility to cephalosporins was near 90%, but they detected a high percentage of staphylococci susceptible to clindamycin (about 80%), ampicillin (about 90%), and gentamicin (about 90%), too [20]. All these findings were in contrast with our data. de Menezes at al. performed a survey in Brazil about bacteria isolated from canine-infected sites; regarding staphylococci, in contrast with our study, 10% of CoPS and 35% of CoNS were resistant to amikacin and 46% of CoPS and 33% of CoNS were resistant to clindamycin; on the other hand, in line with our data, 25% of CoPS and 33% of CoNS were resistant to gentamicin, and the resistance to the several cephalosporins tested ranged from 12% to 38% while considering both CoPS and CoNS [19]. Finally, Gómez-Beltrán and colleagues reported that *S. aureus*, *S. intermedius*, *S. pseudointermedius,* and CoNS isolates from dogs living in Colombia had sensitivities to amikacin ranging from 78% to 100%, to cephalexin ranging from 73% to 83%, and to tetracycline ranging from 38% to 62%, which was in agreement with our data; however, they also detected high levels of susceptibility for gentamicin (59–80%) and ampicillin (74–77%), in contrast to results obtained in our investigation [18].

## 5. Conclusions

Even though *Staphylococcus* spp. are considered to be opportunistic bacteria in dogs, they can infect these animals in different clinical forms. Involved strains have several virulence factors, among which biofilm and gelatinase production seems to be the most frequent.

The present study confirmed *S. pseudointermedius* as the main species involved in canine staphylococcal infections, although other species were often cultured.

Staphylococcal infections are frequently complicated by the antibiotic resistance of the involved strains, which causes difficulties for animal therapy. A large number of the investigated isolates were MDR, even though methicillin resistance did not seem widely spread among them. This variability within the tested isolates confirmed the impossibility to find universally valid antimicrobials and highlighted the importance of in vitro antimicrobial resistance evaluation before antibiotic therapy. In view of the One Health perspective, canine staphylococcal infections should be managed with attention because the same strains could infect humans who live in close contact with dogs. Persons, mainly children, the elderly, and the immunocompromised, could develop severe pathologies that are difficult to treat.

## Figures and Tables

**Table 1 microorganisms-09-00515-t001:** Distribution of *Staphylococcus* species in relation to the site of infection.

Species	N°	Sample Number (and Percentage)				
		Skin	Ear	Nose	SSI	Urine
*Staphylococcus aureus*	6	3 (50.0%)	1 (16.7%)	0	2 (33.3%)	0
*Staphylococcus pseudointermedius*	26	8 (30.8%)	7 (26.9%)	2 (7.7%)	4 (15.4%)	5 (19.2%)
*Staphylococcus xylosus*	10	6 (60.0%)	2 (20.0%)	0	0	2 (20.0%)
*Staphylococcus chromogenes*	4	2 (50.0%)	2 (50.0%)	0	0	0
*Staphylococcus hyicus*	1	1 (100.0%)	0	0	0	0
*Staphylococcus simulans*	1	0	0	0	1 (100.0%)	0
*Staphylococcus haemolyticus*	1	0	0	0	0	1 (100.0%)
*Staphylococcus capitis*	1	0	1 (100.0%)	0	0	0
Total	50	20 (40.0%)	13 (26.0%)	2 (4.0%)	7 (14.0%)	8 (16.0%)

**Table 2 microorganisms-09-00515-t002:** Results of the phenotypic and genotypic tests performed on the examined *Staphylococcus* isolates.

Species	N°	Hemolysis			Phenotypic Characteristic					Virulence Genes					Methicillin Resistance			MDR
		α	β	α/β	Pr	Li	Ge	Sl	Bi	*lukS/F-PV*	*eta*	*etb*	*tsst*	*icaAD*	*mecA*	*mecC*	Ph	
*S. aureus*	6	4	1	1	3	4	4	6	6	0	0	0	1	1	1	0	1	4
*S. pseudointermedius*	26	1	9	16	11	10	24	22	23	0	0	0	0	0	9	0	11	19
*S. xylosus*	10	4	2	4	4	9	5	4	9	0	0	0	0	0	1	0	6	10
*S. chromogenes*	4	1	0	3	3	2	4	3	3	0	0	0	0	0	1	0	1	2
*S. hyicus*	1	0	0	1	1	0	1	1	1	0	0	0	0	0	0	0	0	1
*S. simulans*	1	0	0	1	0	0	1	1	1	0	0	0	0	1	0	0	1	1
*S. haemolyticus*	1	0	1	0	0	0	0	0	0	0	0	00	0	0	0	0	1	1
*S. capitis*	1	0	0	1	1	1	1	1	0	0	0	0	0	0	0	0	1	0
Total	50	10	13	27	23	26	40	38	43	0	0	0	1	2	12	0	22	38

Legend: Pr = protease; Li= lipase; Gel = gelatinase; Sl = slime; Bi = biofilm; Ph = phenotypic resistance detected with oxacillin MIC test; MDR = multidrug-resistant isolates.

**Table 3 microorganisms-09-00515-t003:** Antimicrobial resistance detected in the examined *Staphylococcus* isolates.

Antimicrobials		Number (and Percentage) of Isolates		
Class	Agent	Susceptible	Intermediate	Resistant
Penicillins	AMP	7 (14%)	2 (4%)	41 (82%)
	AML	6 (12%)	2 (4%)	42 (84%)
	AMC	30 (60%)	1 (2%)	19 (38%)
Cephems	CL	32 (64%)	3 (6%)	15 (30%)
	KF	34 (68%)	1 (2%)	15 (30%)
	CTX	33 (66%)	3 (6%)	14 (28%)
	CAZ	26 (52%)	4 (8%)	20 (40%)
Fluoroquinolones	ENR	19 (38%)	7 (14.0%)	24 (48%)
	CIP	22 (44%)	4 8(%)	24 (48%)
Aminoglycosides	CN	27 (54)	5 (10%)	18 (36%)
	S	13 (26%)	5 (10%)	32 (64%)
	TOB	27 (54%)	5 (10%)	18 (36%)
	N	12 (24%)	7 (14.0%)	31 (62%)
	AK	41 (82%)	6 (12%)	3 (6%)
Tetracyclines	TE	18 (36%)	5 (10%)	27 (54%)
	DO	27 (54%)	8 (16%)	15 (30%)
Folate pathway antagonists	SXT	24 (48%)	2 (4%)	24 (48%)
Macrolides	E	3 (6%)	16 (32%)	31 (62%)
Lincosamides	DA	11 (22%)	5 (10%)	34 (68%)
Ansamycins	RD	40 (80%)	5 (10%)	5 (10%)

Legend: AK = amikacin; AMC = amoxycillin and clavulanic acid; AML = amoxycillin; AMP = ampicillin; CL = cefalexin; KF = cephalothin; CTX = cefotaxime; CAZ = ceftazidime; CIP = ciprofloxacin; DA = clindamycin; DO = doxycycline; ENR = enrofloxacin; E = erythromycin; CN = gentamicin; N = neomycin; RD = rifampicin; S = streptomycin; SXT = trimethoprim–sulfamethoxazole; TE = tetracycline; and TOB = tobramycin.

## Data Availability

The data presented in this study are contained within the article.

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
