# Peer review of "Virulence and Antimicrobial Resistance in Canine Staphylococcus spp. Isolates"

_microorganisms, 2021, doi:10.3390/microorganisms9030515_

Round 1

Reviewer 1 Report

General Considerations: In the manuscript “Virulence and antimicrobial resistance in canine Staphylococcus spp. isolates” the authors performed a phenotypic and molecular characterization of 50 clinical Staphylococcus isolates cultured from sick dogs. Considering the increasing prevalence of antibiotic resistant strains and the pathogenic potential of some Staphylococcus species, this kind of studies is important since they can help to evaluate the role of dogs as reservoirs hosts or sources of zoonotic staphylococci. The manuscript is objective, and it is generally well written and clearly presented. However, there are some questions and/or suggestions that I would like to ask.

Minor Comments:

  1. Problems with page numbering due to landscape page.
  2. Line 8: It is not clear what you mean by “could develop different clinical forms”.
  3. Line 66: Instead of “con” it should be “can”.
  4. Line 108: Instead of “was” it should be “were”.
  5. Line 267: Instead of “his detection” it should be “its detection”.

Major Comments:

  1. During the isolation process, did you consider whether the 50 isolates tested were responsible for the infections manifested or if other potentially pathogenic agents could be present?
  2. Materials and Methods Section (2.2 Phenotypic characterization): Could you please describe the criteria used to consider the tested isolates as multi drug resistant.
  3. Line 143: For the genotypic characterization it is not clear why you just analyze the virulence genes described. Considering the low representativeness of the selected genes in your staphylococci isolates, wouldn't it be advantageous to analyze the presence of other virulence genes in these isolates?
  4. Although the results are clear, for the same isolate it is difficult to see whether there is a relationship between the various phenotypes tested, the presence of virulence genes and resistance to antibiotics. I would like to suggest that you detail these data for the 50 isolates creating a supplementary table.
  5. Line 256 and Line 269: The information seems to be contradictory. In line 269, did you mean “the high number of isolates that produce biofilms”?
  6. Line 303-305: Considering that the studies referenced are mostly comparable to the present study, a more detailed analysis could be made in an attempt to perceive these similarities / differences.

Author Response

Reviewer 1

General Considerations: In the manuscript “Virulence and antimicrobial resistance in canine Staphylococcus spp. isolates” the authors performed a phenotypic and molecular characterization of 50 clinical Staphylococcus isolates cultured from sick dogs. Considering the increasing prevalence of antibiotic resistant strains and the pathogenic potential of some Staphylococcus species, this kind of studies is important since they can help to evaluate the role of dogs as reservoirs hosts or sources of zoonotic staphylococci. The manuscript is objective, and it is generally well written and clearly presented. However, there are some questions and/or suggestions that I would like to ask.

Authors want to thank Reviewer 1 for these positive comments and for the time She/He spent in the manuscript revision.

We answered to Reviewer 1 comments at the best of our possibilities and we hope it could be enough.

Minor Comments:

Problems with page numbering due to landscape page.

Dear Reviewer 1, we tried to correct this error (I hope it is ok now, but I it is an automatic configuration of word template and I am not sure about the result)

Line 8: It is not clear what you mean by “could develop different clinical forms”.

In accordance with Reviewer 1 comment, we change “different” with “several”, hoping this is a more appropriate word.

Line 66: Instead of “con” it should be “can”.

Authors modified the text as suggested by Reviewer 1.

Line 108: Instead of “was” it should be “were”.

Authors modified the text as suggested by Reviewer 1.

Line 267: Instead of “his detection” it should be “its detection”.

Authors modified the text as suggested by Reviewer 1.

Major Comments:

During the isolation process, did you consider whether the 50 isolates tested were responsible for the infections manifested or if other potentially pathogenic agents could be present?

We assumed that the employed strains were the agents responsible for the clinical presentation; in most cases they are the only one species, showing a vigorous growth, cultured on blood agar from clinical samples; probably (we did not recorded this information from routine analyses) in some cases contaminants could be present, especially from skin or hear samples, but staphylococci were the predominant species, otherwise we did not take in consideration about them. Authors hope this could be a satisfactory explanation and we are grateful if Reviewer 1 could suggest to us how and where we can insert all these information. A short sentence was added at the and of section 2.1.

Materials and Methods Section (2.2 Phenotypic characterization): Could you please describe the criteria used to consider the tested isolates as multi drug resistant.

As suggested by Reviewer 1, Authors added a description about the criteria used to consider a isolate MDR, immediately after disk diffusion method description; a reference was provided.

Line 143: For the genotypic characterization it is not clear why you just analyze the virulence genes described. Considering the low representativeness of the selected genes in your staphylococci isolates, wouldn't it be advantageous to analyze the presence of other virulence genes in these isolates?

Authors strongly agree with Reviewer 1. However, staphylococci have a lot of virulence genes, some of them found, or sometime searched, only in some species. Authors selected the set of genes in advance, on the basis different criteria: recent publications, site of infection (most of our isolates come from skin and hears), isolated species, zoonotic implications and available resources. Unfortunately, although it would be interesting to expand the set of investigated genes, it was not possible in particular for time and financial reasons. Authors insert a sentence to stress this limitation in discussion section.

Although the results are clear, for the same isolate it is difficult to see whether there is a relationship between the various phenotypes tested, the presence of virulence genes and resistance to antibiotics. I would like to suggest that you detail these data for the 50 isolates creating a supplementary table.

Authors appreciate Reviewer 1 suggestion. Raw data were added as supplementary material.

Line 256 and Line 269: The information seems to be contradictory. In line 269, did you mean “the high number of isolates that produce biofilms”?

Reviewer 1 observation is right, Authors modified line 269 as suggested.

Line 303-305: Considering that the studies referenced are mostly comparable to the present study, a more detailed analysis could be made in an attempt to perceive these similarities / differences.

A real complete confrontation is very hard to perform, considering the different panels of antibiotic employed in the different studies and the different grouping and classification level proposed in the studies. However, Authors, following suggestion of Reviewer 1, tried to highlight the main similarities and difference among the present work and the cited references.

Reviewer 2 Report

The manuscript entitled “Virulence and antimicrobial resistance in canine Staphylococcus spp. Isolates” by Fabrizio Bertelloni, Giulia Cagnoli and Valentina Virginia Ebani, involves a study of 50 staphylococci, isolated from sick dogs, regarding phenotypical and genotypical antimicrobial resistance and virulence factors.

The document is very well written and clear, the conclusions are supported by the results, however no significant development in the field was achieved with this work.

Line 66: can not

Line 73: 2.1. Bacterial strains, isolation and identification

Indicate dates between which the collections were performed.

Lines 102-103: Clarify the procedure for the detection of gelatinase. Were the tubes incubated at 4ºC , 1 h, every day or only at the end of 7 days?

Line 152: Include the meaning for SSI. Surgical site infection?

Line 173: Indicate the number of antimicrobials against which an isolate needs to be resistant to be considered a MDR isolate.

Line 193: These data should also be included in table 2

Line 260: The authors mention that a few isolates were tested for the presence of virulence genes. From the methodology and results I had the impression that all 50 isolates were tested for the presence of these gene. Please, verify and clarify.

Author Response

Reviewer 2

The manuscript entitled “Virulence and antimicrobial resistance in canine Staphylococcus spp. Isolates” by Fabrizio Bertelloni, Giulia Cagnoli and Valentina Virginia Ebani, involves a study of 50 staphylococci, isolated from sick dogs, regarding phenotypical and genotypical antimicrobial resistance and virulence factors.

The document is very well written and clear, the conclusions are supported by the results, however no significant development in the field was achieved with this work.

Authors appreciate the sincere evaluation of Reviewer 2 and the time She/He spent in the manuscript revision. Authors are partially in accordance with Reviewer: with this manuscript we did not discover for the first time something new. However, no recent data from Italy are available; to provide updated information on potentially zoonotic bacteria and antimicrobial resistance, especially in pets, could be important. Furthermore, most studies focused on S. pseudointermedius and S. aureus (all other species if investigated were grouped as CoNS); here detailed information on other species were provided and the importance that these species could have in dog infection was stressed.  

We answered to Reviewer 2 comment at the best of our possibilities and we hope it could be enough.

Line 66: can not

Authors modified the text as suggested by Reviewer 2.

Line 73: 2.1. Bacterial strains, isolation and identification

Indicate dates between which the collections were performed.

Reviewer 2 is right, this is a deficiency. We start to collect strain at the beginning of 2019 and we stopped in March 2020 due to COVID. Information about time of sampling were added in the manuscript.

Lines 102-103: Clarify the procedure for the detection of gelatinase. Were the tubes incubated at 4ºC , 1 h, every day or only at the end of 7 days?

Tubes were incubated at 4°c after 1 week of incubation at 30°C. As suggested by Reviewer 2, this sentence was modified.

Line 152: Include the meaning for SSI. Surgical site infection?

Authors modified the text as suggested by Reviewer 2.

Line 173: Indicate the number of antimicrobials against which an isolate needs to be resistant to be considered a MDR isolate.

In accordance to Reviewer 1 and Reviewer 2 request, this information was added in Material and Method section; if Reviewer 2 think it is necessary to repeat this definition in Table 2 legend, Authors will proceed with this modification too.

Line 193: These data should also be included in table 2

Authors modified Table 2 as suggested by Reviewer 2.

Line 260: The authors mention that a few isolates were tested for the presence of virulence genes. From the methodology and results I had the impression that all 50 isolates were tested for the presence of these gene. Please, verify and clarify.

All isolates were tested for all virulence genes and for all the other characteristics. Authors are sorry for this misunderstanding. The sentence was modified as suggest by Reviewer 2.

Round 2

Reviewer 1 Report

Dear authors,

Globally, I am satisfied with the clarifications and the revision made on the manuscript.

However, I would like to clarify the following comment: “Line 303-305: Considering that the studies referenced are mostly comparable to the present study, a more detailed analysis could be made in an attempt to perceive these similarities / differences.”. In this comment, my point was just to briefly clarify how discrepant / similar the antibiotic resistance of these isolates is compared to those of the studies mentioned. In fact, in most of the studies mentioned, most isolates (albeit in somewhat variable percentages) are resistant to amikacin, rifampicin, and even cephalosporins. Perhaps the most marked difference resides in resistance to B-lactams. Perhaps, I did not explain myself properly, but my suggestion was more related to an analysis of this type.

Minor Comments:

Line 320: Instead of "a very variable levels" --> "variable levels"

Author Response

Reviewer 1 comments

Dear authors,

Globally, I am satisfied with the clarifications and the revision made on the manuscript.

Authors are pleasured that Reviewer 1 appreciated our modifications. We tried to improved again the manuscript following Reviewer 1 suggestions.

However, I would like to clarify the following comment: “Line 303-305: Considering that the studies referenced are mostly comparable to the present study, a more detailed analysis could be made in an attempt to perceive these similarities / differences.”. In this comment, my point was just to briefly clarify how discrepant / similar the antibiotic resistance of these isolates is compared to those of the studies mentioned. In fact, in most of the studies mentioned, most isolates (albeit in somewhat variable percentages) are resistant to amikacin, rifampicin, and even cephalosporins. Perhaps the most marked difference resides in resistance to B-lactams. Perhaps, I did not explain myself properly, but my suggestion was more related to an analysis of this type.

As suggested by Reviever 1, the paragraph between line 315 and line 340 was modified, highlighted for each of the reported study the main similarity and differences with our study.

Minor Comments:

Line 320: Instead of "a very variable levels" --> "variable levels"

The text was modified as suggested by Reviewer 1.